# A User-Specific Approach for Comfortable Application of Advanced 3D CAD/CAM Technique in Dental Environments Using the Harmonic Series Noise Model

**Eun-Sung Song** [1], **Young-Jun Lim** [2],*  and **Bongju Kim** [1]

1   Clinical Translational Research Center for Dental Science, Seoul National University Dental Hospital, Seoul 03080, Korea; songeunsung@gmail.com (E.-S.S.); bjkim016@gmail.com (B.K.)
2   Department of Prosthodontics and Dental Research Institute, School of Dentistry, Seoul National University, Seoul 03080, Korea
*   Correspondence: limdds@snu.ac.kr; Tel.: +82-02-2072-2940

**Abstract:** Recently, there has been a focus on improving the user's emotional state by providing high-quality sound beyond noise reduction against industrial product noise. Three-dimensional computer aided design and computer-aided manufacturing (3D CAD/CAM) dental milling machines are a major source of industrial product noise in the dental environment. Here, we propose a noise-control method to improve the sound quality in the dental environment. Our main goals are to analyze the acoustic characteristics of the sounds generated from the dental milling machine, to control the noise by active noise control, and to improve the sound quality of the residual noise by synthesized new sound. In our previous study, we demonstrated noise reduction in dental milling machines through tactile transducers. To improve the sound quality on residual noise, we performed frequency analysis, and synthesized sound similarly as musical instruments, using the harmonic series noise model. Our data suggest that noise improvement through synthesis may prove to be a useful tool in the development of dental devices.

**Keywords:** 3D CAD/CAM dental milling machine; noise control; sound quality; sound synthesis; harmonic series

## 1. Introduction

Technological advancement has always benefitted the medical field, especially in the enhancement of the medical environment. Recent amelioration in algorithmic science allows us to simulate and solve many 'real system' problems, previously inaccessible due to practical limitations such as including unknown dynamics and large or continuous state, action, and parameter spaces in data poor environments [1,2]. 'Noise' is one of the real system problems currently accessible through this method.

Computer aided design and computer-aided manufacturing (CAD/CAM) milling has proven to be advantageous in various fields, through improved product design, higher productivity, better utilization, and stronger quality control [3]. Advanced CAD/CAM technologies have actualized precise and patient-customized prosthesis in the field of dentistry [4]. However, the 3D CAD/CAM milling machines work as a source of disruptive noise, as they add quite a cacophony to the dental environment. This has been acknowledged as an even greater problem [5,6].

It is evident that stress and disruption causing external sounds in the dental environment should be regulated [7]. Currently, there are limited amount of research on noise reduction in dental clinics, and a serious lack of research on modification and qualitative aspects of noise in dental environments.

Our previous study [8] demonstrated that noise reduction of CAD/CAM dental milling machines can be done through an active vibration control (AVC) method with tactile transducers.

Earlier studies accomplished noise reduction through active noise control methods [9]. Moreover, a recent study identified 'residual noise', after noise reduction, as an important key in increased auditory comfortability, and the consumer's emotional quality [10].

Previous literature by Genuit reported a reduction in unwanted noise through technical development [11]. However, they considered the complete removal of such noise from machines, as impossible. It is notable that Luigi R. et al. suggested that any kind of noise can be used as music as a consequence of more adaptation of noises using technologies [12].

The size of the noise felt is not solely determined by the sound pressure level, and thence it is necessary to focus on changing the quality in terms of auditory characteristics [13]. A product's noise is an important factor for consideration in relation to user's convenience. This all comes down to a conclusion that attenuation of noises, as well as converting noises generated from CAD/CAM milling machines into more subjectively suitable sounds, is a crucial factor for the well-being of the dental environment.

In order to improve this problem, we propose a noise improvement method through sound source characterization and sound control technology. Our study focuses on objective criteria that can make a rasping sound into a sound pleasant to the ear. We also focus on sound enhancement through tone analysis of instruments and structural understanding of harmonics. Our approach is expected to contribute the noise quality improvement process as real-time sound quality enhancement systems.

## 2. Materials and Methods

In our previous study [8], we obtained noise reduction of a CAD/CAM dental milling machine through the vibration control method. However, those results just indicated a decrease in noise levels. Sound frequency disruptiveness, or the subjective level of discomfort with a specific noise, varies from person to person. Therefore, in this study, further experimentation was conducted to improve the sound quality of the remaining noise based on the noise reduction results obtained in the previous study. A description block diagram of this study is shown in Figure 1.

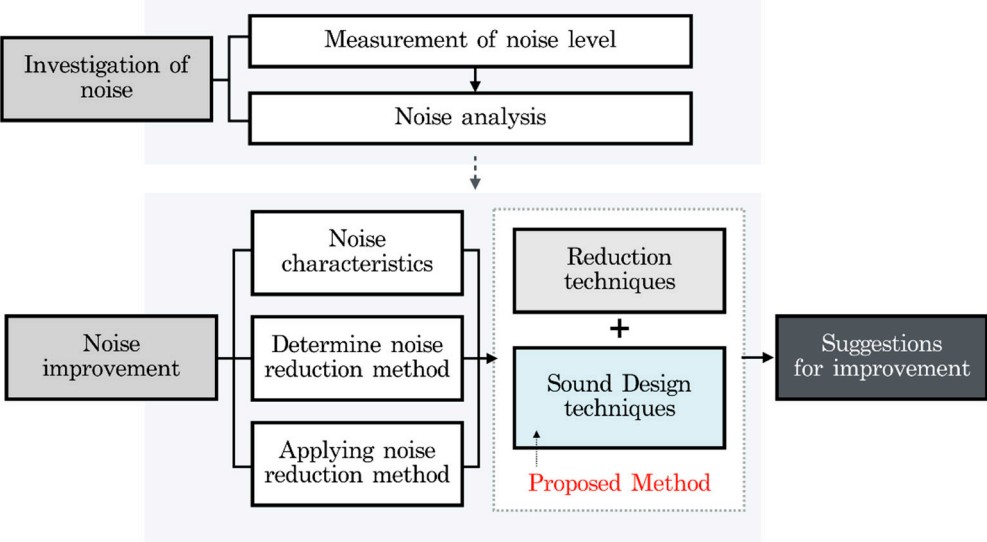

**Figure 1.** Noise improvement process.

## 2.1. Measurement of Noise

The CAMeleon CS CAD/CAM milling machine (NeoBiotech, Seoul, Korea) was employed to obtain all test results presented in this study. The noise level was measured while the instrument was running with a diamond bur cutting a zirconia block. Noise was analyzed by sound pressure level and frequency spectrum. The sound level intensity was measured with a digital sound level meter (Lutron Sl-4013®, Lutron, Taipei, Taiwan) by using the dB(A) scale. The frequency spectrum of noise was investigated by a half-inch microphone (ECM8000®, Behringer, Willich, Germany). Initially, measurements of acoustic noise level were performed for 5 min while the dust collector was active. Afterwards, to analyze the characteristics of the noise of the dust collector, the noise signal of the dust collector was again recorded using a microphone. The recording microphone and sound level meter were located horizontally a distance of 30 cm away from position A (Figure 2).

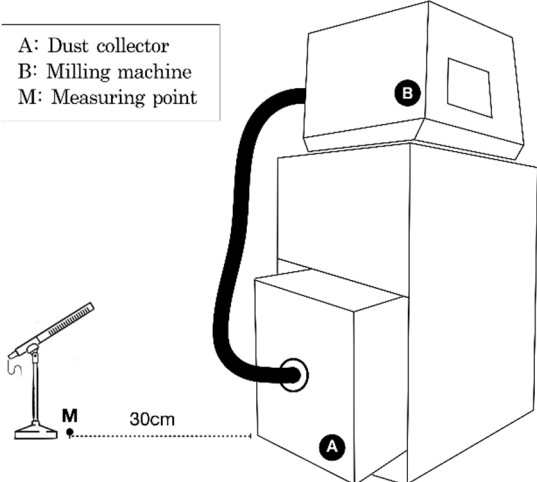

**Figure 2.** Location of recording microphone and sound level meter.

## 2.2. Sound Synthesis

The data for sound design were created using the *open Frameworks* program (an open source toolkit designed for creative coding, written in C++ and built on top of OpenGL), an environment and programming language. The sound was produced using a virtual studio technology instrument (VSTi) and musical instrument digital interface (MIDI) virtual instruments. MIDI communication was handled over open sound control (OSC), a protocol for networking sound synthesizers, computers, and other multimedia devices for purposes such as musical performance.

## 3. Noise Improvement Plan

### 3.1. Analysis of Noise

The results of the active cancellation procedure in our previous study [8] are shown in Table 1. Test results showed that the primary source of noise, dust collector vibrations, was effectively decreased through the anti-vibrations produced by the AVC method. An approximately 11 dB(A) decrease in intensity of sound pressure was observed after implementation of the AVC method. In graph C of Table 1, which shows each result, the fast Fourier transform (FFT) analysis enlarges the frequency domain by log scale. Graph B shows the frequency components such as partials, overtones, and harmonics as a linear scale at a glance through FFT analysis.

**Table 1.** Before and after dust collector noise level control: Noise frequency components ((**A**) Size of each harmonic components, (**B**) fundamental frequency and overtones, and (**C**) fast Fourier transform (FFT) analysis spectrum) and results of noise level ((**D**) average SPL (sound pressure level)).

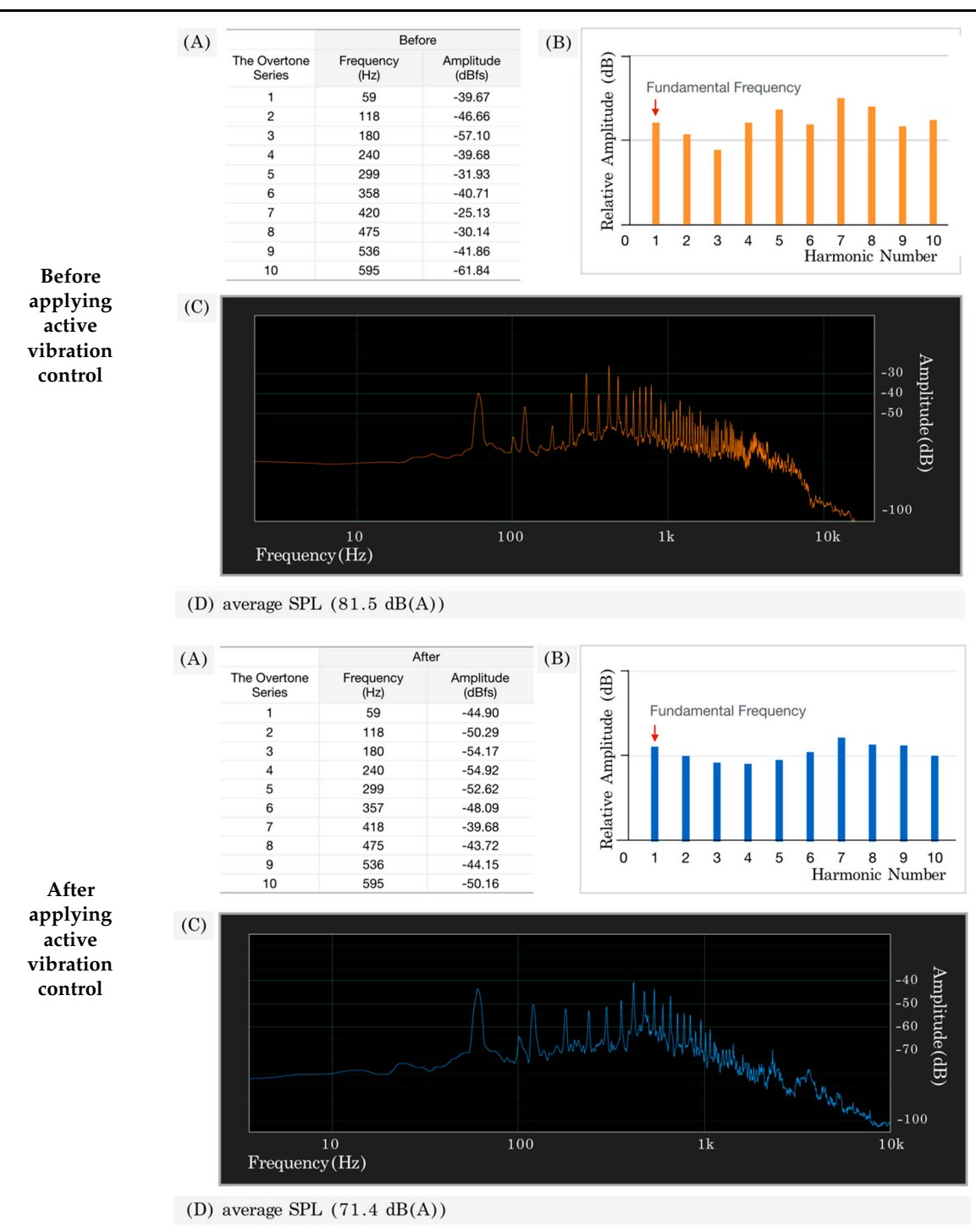

The noises produced by the milling machine dust collector are discrete tones with a fundamental sound frequency of about 60 kHz and periodic sound pressure fluctuations that bring about a sense of pitch consisting of harmonics of its multiple. After applying the AVC (see Table 1), frequency analysis of the spectral data indicated that the strongest energy was found to be concentrated in the low-frequency region. High-frequency components of harmonics were relatively greater with lower

amounts of frequency control. A complex tone, which is a sound with a timbre particular to the instrument, is composed of the lowest harmonic and its integer multiples [14]. As shown in Table 1, the frequency of a noise with a line spectrum of the characteristics of a milling machine has similarities to the timbre and harmonic structure of a general musical instrument. Therefore, we concluded that it would be possible to design a sound associated with this noise that contained musical qualities.

*3.2. Sound Design*

In the previous study, the AVC method indicated a decrease in noise levels, and we proposed a method focusing on maximum attenuation of the primary source of noise. However, because the AVC method cannot be applied in real time, due to error [15] and noise subjectivity, it is also important to consider enhancing the quality of sound as well. Sound quality can be improved not only by considering noise reduction, but also a system that applies psychoacoustics, which is concerned with the characteristics of residual noise to match human preference [16]. Several studies [17,18] that linked annoying qualities of a sound to its loudness found significant correlations between annoyance and loudness judgments. Loudness judgments might be affected by the emotional experience. Significant unpleasant stimuli, compared to neutral or positive stimuli [19], might be perceived as louder due to their negative effect [20].

By controlling for these psychological factors, we might be able to generate a more desirable sound. Generally, evaluation of the sound quality index can be done through Zwicker's method [21] of considering auditory systems in terms of loudness, sharpness, roughness, fluctuation strength, etc. It is true that the factor of roughness has been standardized according to the international ISO 532B (ISO 532 B, 1975) regulatory standard [22]. However, the remaining factors have not been standardized; therefore, the results of most sound quality indices differ depending on the analysis method and program [23]. Hence, a method that can be judged more generally after sound quality improvement is needed. Musical tones are generally referred to as good-quality sounds due the harmonic nature of their compositional structure. With the organization of some peak frequency components in their specific compositional realm, harmonics are felt as comfortable sounds. However, previous literature reported that there have been discrepancies among the different sound quality indices. Therefore, we propose reforming residual noises from CAD/CAM milling machines using a harmonic structure of music according to the noise characteristics of the machinery after setting the standard of qualitatively preferred sounds as musical tones. Since musical instruments have unique harmonics, which are the main structure of music and the important factor in determining tone color [24], we tried to produce the most favorable sound by correcting the tone into harmonics similar to musical instruments.

3.2.1. Experimental Method

The first experimental design consisted of a tactile transducer [25] and a control unit to generate anti-vibrations. The tactile transducer included speakers capable of both actively generating and controlling anti-vibrations of a sound source. In this experiment, we used the first experimental design to synthesize the sound. A diagram of the vibration reduction and sound improvement method is shown in Figure 3.

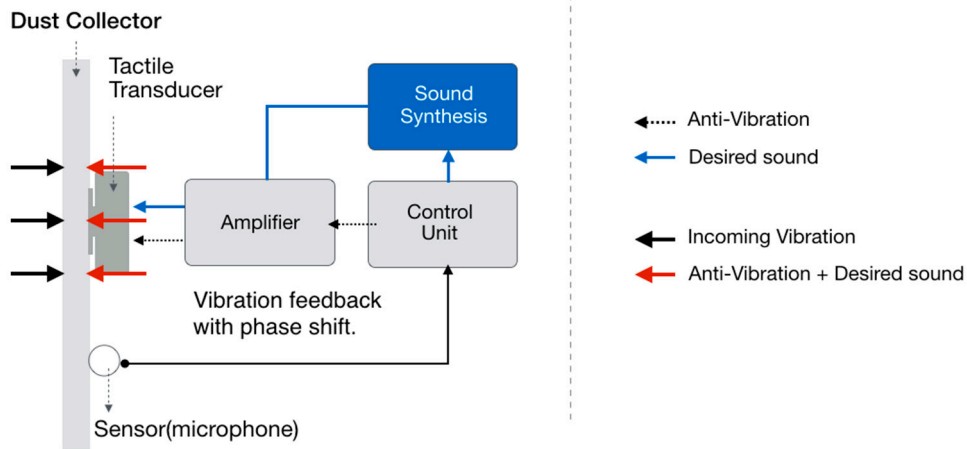

**Figure 3.** Modelling procedure scheme.

### 3.2.2. Harmonic Series Noise Model

A harmonic is any group of harmonic series or an ideal set of frequencies that have a fixed interval with regard to a common fundamental frequency. A musical tone consists of several harmonics, making it a so-called complex tone [26]. Each note's frequency is calculated from the reference sound by Equation (1), and each harmonic integer's fundamental frequency can be calculated by Equation (2):

$$f : f\sqrt[12]{2^1}f : f\sqrt[12]{2^2}f : f\sqrt[12]{2^3}f : f\sqrt[12]{2^4}f : f\sqrt[12]{2^5}f \cdots : f\sqrt[12]{2^{11}}f : 2f \tag{1}$$

$$f : 2f : 3f : 4f : 5f : 6f : 7f : 8f : 9f : 10f : 11f : 12f : 13f : 14f : 15f : 16f \tag{2}$$

The fundamental frequency determines the pitch, and the arrangement of harmonics determines the timbre of the instrument. Every musical instrument has a different harmonic component [27] (Figure 4).

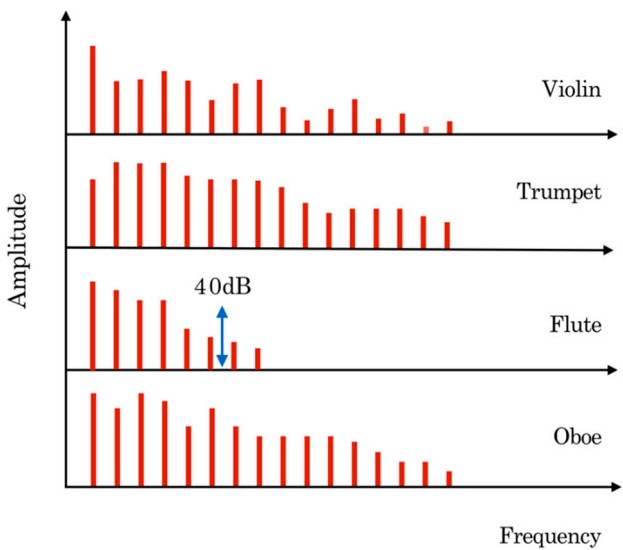

**Figure 4.** Harmonics of four instruments playing A4: 440 Hz.

To improve noise quality, we used a harmonic structure to design sounds that resembled those of musical instruments. By analyzing the structure and energy of the harmonics of musical instruments, we changed the energy and structure of the noises of the milling machine as well, to make those noises

closer to those of the harmonics of musical instruments. This system was designed to not only identify the sounds and noises created by the musical instruments but also to compare and extract components of frequencies that could be used to control and improve the noise.

### 3.2.3. Determination of Sound Design

To analyze the frequencies of the sounds produced by the milling machine, a 16-bit 44.1 kHz noise was digitally converted into frequency data while applying fast Fourier transform (FFT). The frequency spectrum with the strongest energy was extracted and selected as F0, the fundamental frequency. The second and third orders were assigned as F1 and F2, respectively. The spectrum was analyzed with the structure having one independent pitch and timbre constructing the harmonic series. The frequency components of the milling machine sounds after applying the noise-reduction method are shown in Figures 5 and 6. The musical instrument sounds were compared to those of the milling machine dust collector.

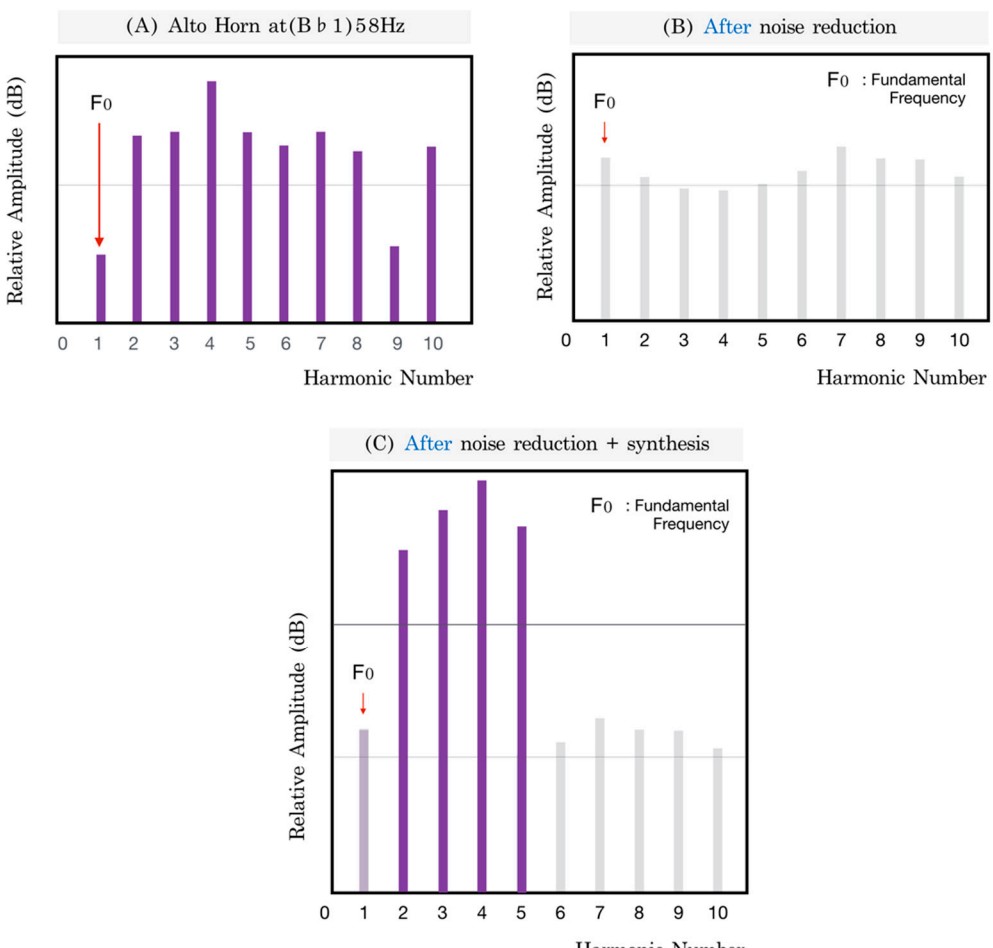

**Figure 5.** Harmonic spectra. (**A**) Alto Horn at B♭1, (**B**) milling machine after noise reduction, and (**C**) the synthesis of alto Horn and milling machine after noise reduction.

A number of wind instruments were selected as candidates, and the harmonic waveform was compared to that of the noise of milling machines. As shown in Figure 5, for example, the alto horn's fundamental harmonic amplitudes in in B♭1 (about 58 Hz) is at least 5 dB weaker than its following harmonics. If the sound is synthesized similar to horn instruments, it can produce louder noises, as shown in Figure 5C. For this reason, when choosing a musical instrument to refer to the harmonic structure, the following two conditions are considered as priorities: First, the strongest

noise was generated at 60 Hz in the milling machine, which showed the greatest decrease after noise control. We selected the musical instrument that had a harmonic structure of overtone greater than the fundamental frequency in order to reduce the effect on the noise after 60 Hz synthesis. Second, we decided to target instruments that could have a similar harmonic structure, except the fundamental frequency, when reconstructing harmonics. Compared to organ pipe harmonics [28] in Figure 6, milling machine harmonics were found to be best suited for making similar in their composition. Therefore, the organ pipe harmonics were used as a musical object to compare to the harmonics of the milling machine dust collector. As a result, as shown in Figure 6, the flue organ pipe has the most suitable structure, and its harmonics were used for musical comparison with the harmonics of the milling machine dust collector. The harmonics of the milling machine frequency bands with insufficient energy were compensated through the design and synthesis of musical instrument sounds. The frequency bands of the additionally designed sounds used in the synthesis were 118 Hz, 180 Hz, 240 Hz, and 299 Hz (see Figure 7). The harmonic structure of the pipe organ was synthesized to appear at the same rate. These sounds were designed to minimize the effects of sound pressure when anti-vibration motion was activated. An overview of the proposed method in this study is shown in Figure 8 as a block diagram.

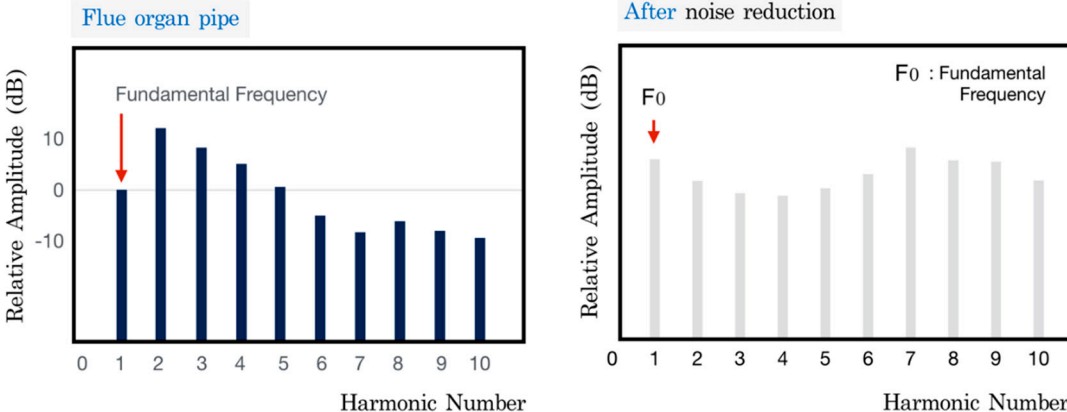

**Figure 6.** Components and sizes of milling machine frequencies vs. harmonic structure of flue organ pipe (Malvern Priory, UK).

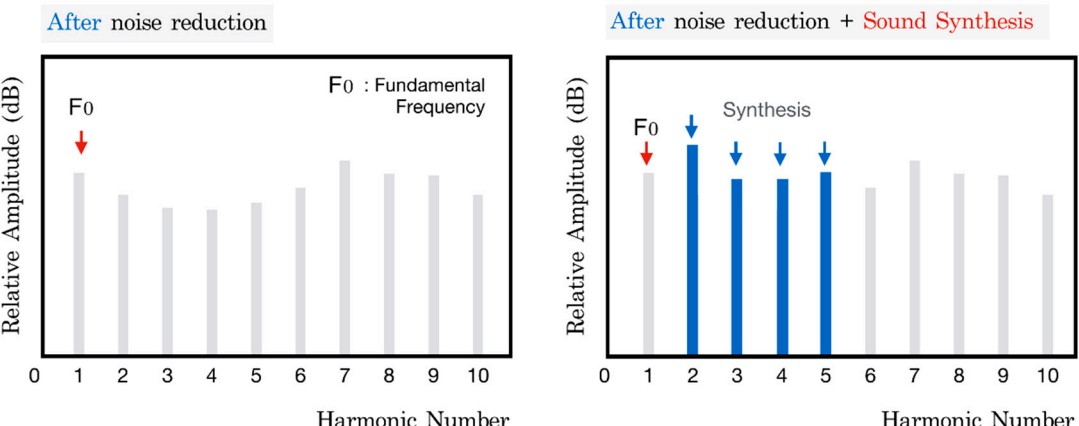

**Figure 7.** Frequency components compensated by harmonic synthesis.

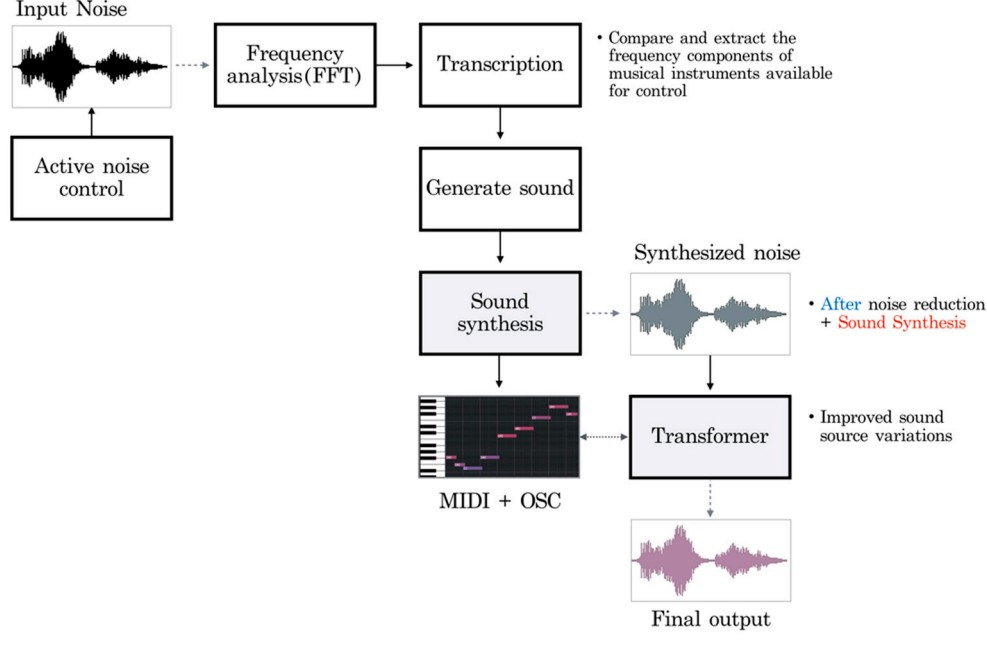

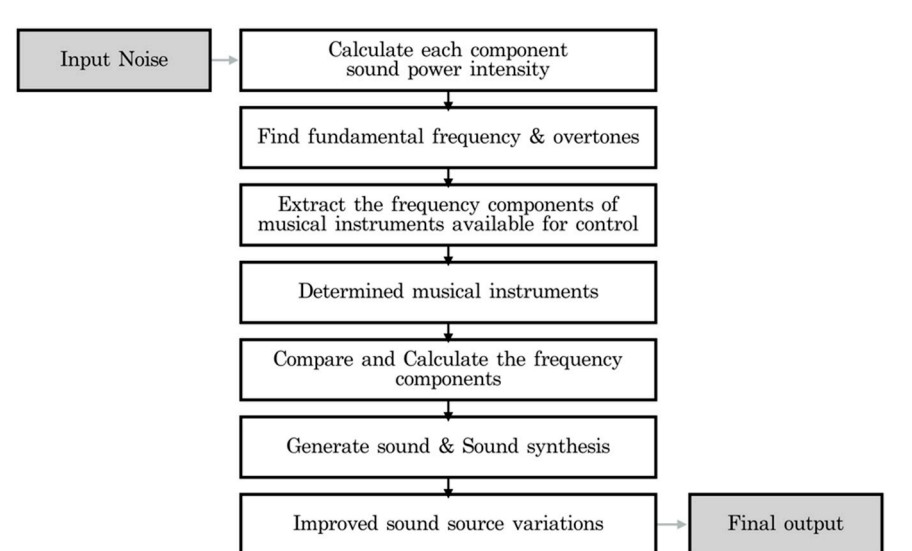

**Figure 8.** Overview of proposed method in this study.

## 4. Result

After synthesizing the designed sound as that of a milling machine, the corrected noise spectrum and frequency analysis results were recorded, as seen in Figures 9 and 10. By comparing the frequency components of the selected musical instrument and the milling machine (see Figure 6), unlike the evenly distributed spectra of uncorrected everyday noises, the corrected noise spectrum's whole frequency band had a clear line spectrum. However, the results also indicated the presence of sounds that seemed to be noises other than those of the musical frequency components. Table 2 shows similar results of noise levels (dB(A)) before and after sound synthesis. The newly redesigned noise can reproduce a similar tone with a harmonic structure like the musical instrument, but this is only a consideration of tone, which is one of several musical units. In order for it to be a form of music that serves as a standard for good sound, we must consider various elements of music such as melody, harmony, and rhythm. These organizational elements express a musical idea but lack sufficient weight

to stand alone [29]. We therefore present an additional example of a sound source with a minimal musical form through noise changes as a final output (see Figure 11).

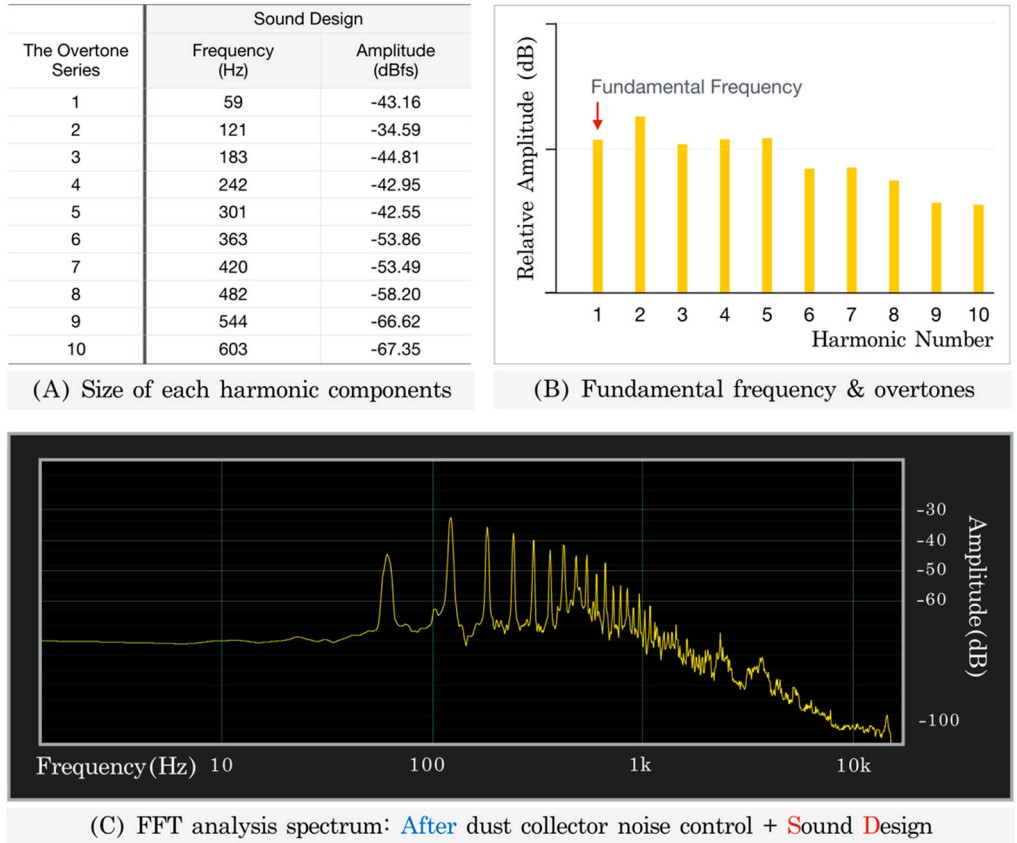

(A) Size of each harmonic components

(B) Fundamental frequency & overtones

(C) FFT analysis spectrum: After dust collector noise control + Sound Design

**Figure 9.** (**A**) Size of remaining harmonic components after compensation of frequency; (**B**,**C**) frequency spectrum analysis after synthesis and compensation.

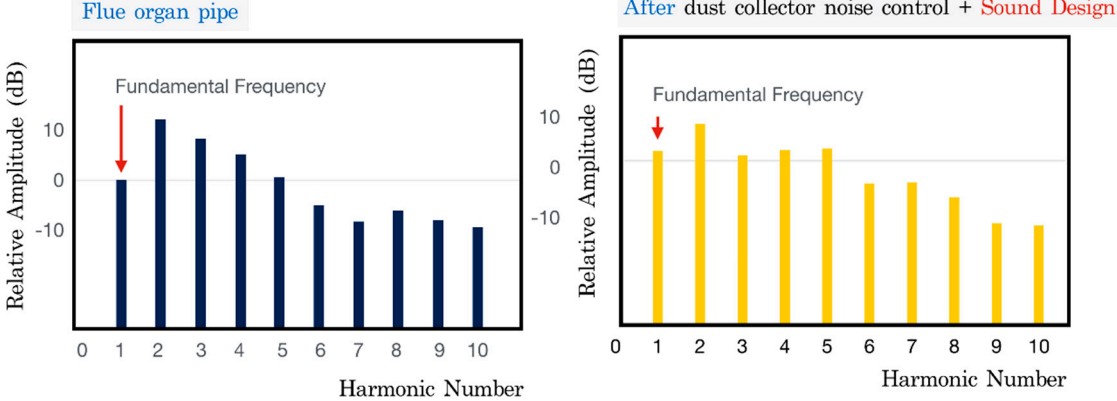

**Figure 10.** Frequency components of the selected musical instrument and the designed milling machine sound.

**Table 2.** Results of noise levels (dB(A)) before and after sound synthesis.

| Average | Before Synthesis | After Synthesis |
| --- | --- | --- |
| SPL (dB(A)) | 78.3 | 79.62 |

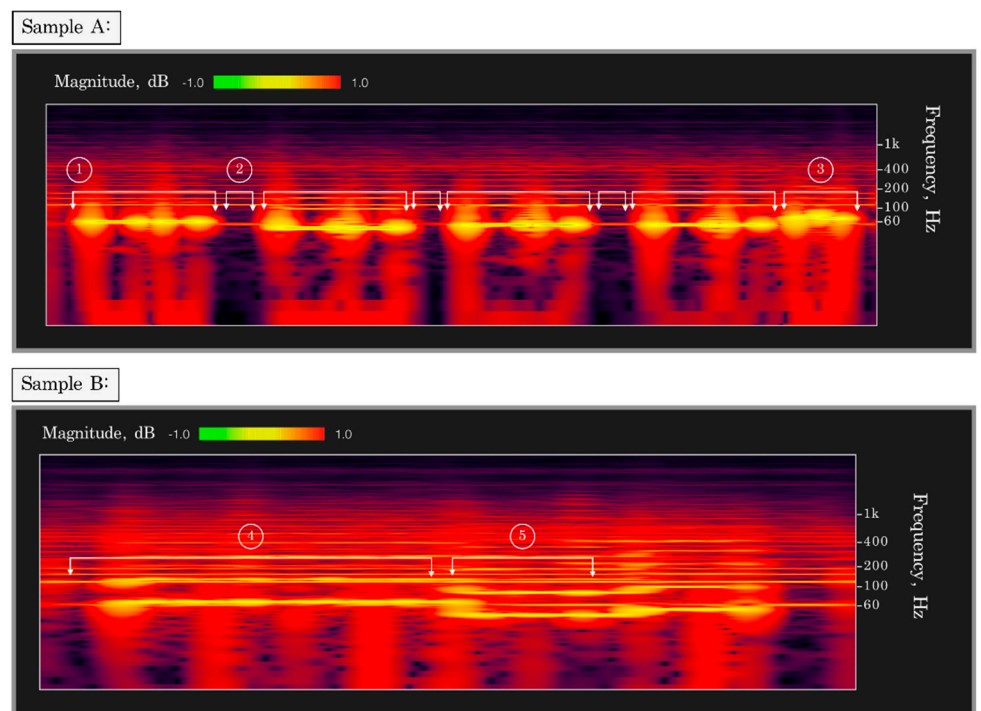

**Figure 11.** Spectrograms of sample sources A and B from designed milling machine sound using virtual instruments. Numbers 1 to 5 indicate sections that were modified to have a musical form that is not a constant note.

Figure 11 shows spectrograms of sample sources A and B, designed as the final output. The sound source was produced using VSTi and MIDI virtual instruments. The spectrograms show the frequency band and sound intensity of the entire sound source. The change in frequency means that the pitch of the sound is changing, and the intensity of the sound is changed by the amplitude (dB) value, which shows that there are various changes in the monotone sound. The graph is shown on a logarithmic scale to show the magnification of the frequency domain.

## 5. Discussion and Conclusions

Similar to general noises, CAD/CAM milling machines produce many frequencies of similar sizes, which causes the production of rough nonharmonic tones that lack sound matching. Due to similarities between the harmonic structures of principal organ pipes and the reduced noise of milling machines, we modeled sound improvement of the reduced noise after this instrument's frequency spectrum and fundamental frequency. In addition, we showed the possibility of various musical sound sources by designing the sound tones by changing them using minimal musical elements. As a result, we constructed a harmonic structure using the reduced noise by adding harmonic numbers that were missing. The frequency bands that were additionally synthesized to resemble the organ pipe harmonics were the second to fourth harmonic numbers. Harmonics is a characteristic of sound that distinguishes sounds in terms of pitch, loudness, spectrum, and envelope. However, because most acoustic instruments produce prominent formant frequencies, in order to successfully control for quality of noise, it is also important that we consider formant properties of individual instruments. In addition, improving the post-enhancement algorithm can be considered as a possibility in developing a sound quality method that is more effective at rearranging the composition of the attenuated noises into the sounds of musical consonance.

The results from our initial experiment indicated a reduction of the target noise through the use of anti-vibrations produced by AVC. In addition to an AVC method, to further improve the environmental noise of dental clinics, we focused on integrating the harmonic series noise model to increase the level

of qualitative acoustic comfort. This study identifies excellent sound using the tone color of musical instruments, and proposes a method that makes these harmonics similar for the improvement of tone color. Our result indicated that the tones are corrected by synthesis, and a change of multiple musical factors of tone has shown the possibility for musical transformation of the original sound source. The tone, which is a product of synthesis, is difficult to assess, subjectively or objectively. Therefore, we assessed it by making similar harmonics, selecting a musical instrument with a similar tone color to avoid louder noise, and reflecting the most desirable options.

The sound source of tone to be a basis will be established by clinical assessment. Therefore, further studies need to quantitatively analyze the musical form of sound by improving algorithms and measuring the psychological status of subjects. It is difficult to assess an individual's psychological state objectively through the general questionnaire-based method. Recent studies suggested long-term mental health monitoring using the wearable devices [30,31]. It is expected that the developed algorithm will be improved by objective and continuous feedback from long-term monitoring methods in future studies. Additionally, repetition of the same musical composition could be disruptive. In order to improve on the proposed model of quality control, further research could focus on developing an algorithm that makes the music change at certain time periods based on the noise being produced at the time. One of the possible methods is that tone extraction algorithms, such as mel-frequency cepstrum coefficient (MFCC) feature-based machine learning algorithms, can automatically sort out appropriate tones from machine noises in real-time [32]. It is not a far-fetched expectation that user-specific comfortable sound sources can be provided through the application of machine learning in the future.

**Author Contributions:** Conceptualization, E.-S.S.; methodology, Y.-J.L.; software, E.-S.S.; validation, B.K.; data curation, E.-S.S. and B.K.; writing—original draft preparation, E.-S.S.; writing—review and editing, B.K. and Y.-J.L.; visualization, E.-S.S.; supervision, Y.-J.L.

**Funding:** This work was supported by grant no. 04-2019-0103 from the Seoul National University Dental Hospital Research Fund, Seoul National University Dental Hospital. The APC was funded by Dental Research Institute, School of Dentistry, Seoul National University, Seoul, Republic of Korea.

**Conflicts of Interest:** The authors declare no conflict of interest.

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
