# Peer review of "A User-Specific Approach for Comfortable Application of Advanced 3D CAD/CAM Technique in Dental Environments Using the Harmonic Series Noise Model"

_applsci, doi:10.3390/app9204307_

Round 1
Reviewer 1 Report
Thank you for your submission. This was a very interesting manuscript and overall well written.
I have a few comments.
Introduction:
You clearly state your purpose in line 44-46. this is important, thank you for doing that.Methods:
The methods section is somewhat confusing. you label it at section 2 with measurement of noise as a subheading on line 56 and sound synthesis on line 70 - should these be 2.1 and 2.2? Is the Noise Improvement plan part of the methods? There is 3.1 on line 77 and 3.2 on line 100 but then there is another 3.1 on line 126. Please relabel as appropriate you headings - if this is part of the methods section please clearly state that. In the Noise Improvment Plan section - have table 1. Please include what "A", "B", and "C" are defined as in the legend, as the table should be able to stand on its own if a reader looked at it.Results:
There is no clearly defined Results section. This manuscript is describing a method for noise reduction through harmonics - but it would be helpful to the reader to have a clear "Results" section to summarize what you found in through your methods.I think it is key that you bring up machine learning in line 243-244. It would be very interesting to see what an algorithm would do to reduce noice in the CAD/CAM machines - if it woudl pick the same harmonics as you did - the flue pipe organ.
Author Response
Dear Reviewer
Thank you for your kind comment.
Please confirm the attached file.
Sincerely yours,

Reviewer 2 Report
The paper analyze and modify the characteristics of noise by observing and measuring the sound generated by a milling machine. It is an interesting topic. However, the organize of the paper is not well, the structure is read as a report rather than a paper. The author need do strong edit and reorganize. In addition, more literature review should be included such as 1.Sihao Yang, Bin Gao, Long Jiang, Jikun Jin, Zhao Gao, Xiaole Ma, and W.L. Woo, IOT structured Long-term Wearable Social Sensing for Mental Wellbeing, IEEE Internet of Things Journal, vol. 6, no. 2, pp. 3652-3662, Apr 2019. 2.Long Jiang, Bin Gao, Jun Gu, Yuanpeng Chen, Zhao Gao, Xiaole Ma, Keith M. Kendrick, and W.L. Woo, Wearable Long-term Social Sensing for Mental Wellbeing, IEEE Sensors Journal, 2018. More results evaluation as well as comparison should be added
Author Response

(The authors gave the same response as above.)

Reviewer 3 Report
The paper discusses an important subject. The paper is well written and organized. My comments are as follows:
1- The introduction needs to be rewritten. It is short.
2- The main contribution of the paper is not well described in the abstract. The authors need to rewrite the abstract by emphasizing the main contribution more clearly.
3- I highly encourage the authors to include some of the recently published papers in their introduction. In particular:
- Textile Application: From Need to Imagination. In Textiles for Advanced Applications. IntechOpen, 2017.
- "Bayesian Control of Large MDPs with Unknown Dynamics in Data-Poor Environments”, Advances in Neural Information Processing Systems, pp. 8146-8156, 2018
- “MFBO-SSM: Multi-Fidelity Bayesian Optimization for Fast Inference in State-Space Models”, AAAI, 2019.
4- Adding a diagram can help the readability of the paper.
5- The format of some the references is not in standard form. These need to be fixed.
Author Response

(The authors gave the same response as above.)

Round 2
Reviewer 2 Report
The revised paper is ok for publish
Reviewer 3 Report
The paper is well-revised and in my opinion it is ready for publication.